# FEDERATED LEARNING WITH OPENSET NOISY LABELS

## ABSTRACT

Federated learning is a learning paradigm that allows the central server to learn from different data sources while keeping the data private at local. Without controlling and monitoring the local data collection process, it is highly likely that the locally available training labels are noisy, just as in a centralized data collection effort. Moreover, different clients may hold samples within different label spaces. The noisy label space is likely to be different from the unobservable clean label space, resulting in openset noisy labels. In this work, we study the challenge of federated learning from clients with openset noisy labels. We observe that many existing solutions, e.g., loss correction, in the noisy label literature cannot achieve their originally claimed effect in local training. A central contribution of this work is to propose an approach that communicates globally randomly selected "contrastive labels" among clients to prevent local models from memorizing the openset noise patterns individually. Randomized label generations are applied during label sharing to facilitate access to the contrastive labels while ensuring differential privacy (DP). Both the DP guarantee and the effectiveness of our approach are theoretically guaranteed. Compared with several baseline methods, our solution shows its efficiency in several public benchmarks and real-world datasets under different noise ratios and noise models.

## 1 INTRODUCTION

With the development of distributed computation, federated learning (FL) emerges as a powerful learning paradigm for its ability to train with data from multiple clients with strong data privacy protection (McMahan et al., 2017; Kairouz et al., 2021; Yang et al., 2019). With each of the distributed clients having a different collection and annotation process, their observed data distributions are likely to be highly heterogeneous and noisy. This paper aims to provide solutions for a practical FL setting where not only do each client's training labels carry different noise rates, the observed label space at these clients will differ as well, even though their underlying clean labels are drawn from the same label space. For example, in a global medical system, the causes (labels) of disease are annotated and reported by doctors, and these labels are potentially noisy due to the differences in the doctors' training backgrounds (Ng et al., 2021). When certain causes and cases can only be found in data clients from country $A$ but not country $B$, the observed noisy label classes in country $A$ will then differ from the one of country $B$. We call such a federated learning system has *openset noise problems* if the observed label space differs across clients.

We observe that the above openset label noise will pose significant challenges if we apply the existing learning with noisy labels solutions locally at each client. For instance, a good number of these existing solutions operate with centralized training data and rely on the design of robust loss functions (Natarajan et al., 2013; Patrini et al., 2017; Ghosh et al., 2017; Zhang & Sabuncu, 2018; Feng et al., 2021; Wei & Liu, 2021; Zhu et al., 2021a). Implementing these approaches often requires assumptions, which are likely to be violated if we directly employ these centralized solutions in a federated learning setting. For example, loss correction is a popular design of robust loss functions (Patrini et al., 2017; Natarajan et al., 2013; Liu & Tao, 2015; Scott, 2015; Jiang et al., 2022), where the key step is to estimate the label noise transition matrix correctly (Bae et al., 2022; Zhang et al., 2021b; Zhu et al., 2021b; 2022). Correctly estimating the label noise transition matrix requires observing the full label space, when the ground-truth labels are not available. In FL where the transition matrix is often estimated only with the local openset noisy labels, existing estimators

of the noise transition matrix would fail. Moreover, even though we can have the best estimate of the noise transition matrix as if we have the ground-truth labels for the local instances, the missing of some label classes would make the estimate different from the ground-truth one, and again leads to failures (detailed example in Section 3.2).

Given the difficulties in estimating the noise transition matrix, we develop a new solution *FedPeer* to tackle the challenge of learning from openset noisy labels in FL. Our solution is inspired by the idea of using "contrastive labels", whose implementation does not require the knowledge of the noise transition matrix. Notable examples of contrastive labels in the learning with noisy label communities include negative labels (Kim et al., 2019; Wei et al., 2022a), peer labels (Liu & Guo, 2020), and complementary labels (Ishiguro et al., 2022; Feng et al., 2020). The high-level idea is to introduce a negative loss using contrastive labels to punish a model for overfitting to the noisy label distributions. Nonetheless, applications of these approaches would require sampling of a global "contrastive" noisy labels - constructing local contrastive labels in each client will be problematic again since different clients may have different noisy label spaces in the openset noise setting. Our solution FedPeer has an explicit step to communicate labels among clients in a differentially private (Dwork, 2008; Dwork et al., 2014) way.

Our contributions are summarized as follows.

- We formally define the openset noise problem in FL, which is more practical than the existing homogeneous noisy label assumptions. The challenges along with the openset noise are also motivated by analyzing the failure cases of the existing popular noisy learning solutions such as loss correction (Natarajan et al., 2013; Patrini et al., 2017; Liu & Tao, 2015).
- We propose a novel framework, FedPeer, to solve the openset label noise problem. FedPeer builds on the idea of contrastive labels, and adopts peer loss (Liu & Guo, 2020) as a building block.
- To mitigate the gap between centralized usage of contrastive labels and the federated one, we propose a *label communication* algorithm with a differential privacy (DP) guarantee. We also prove that benefiting from label communication, the gradient update of aggregating local peer loss with FedAvg is guaranteed to be the same as the centralized implementation of peer loss, therefore establishing its robustness to label noise.
- We empirically compare FedPeer with several baseline methods on both benchmark datasets and practical scenarios, showing that, in terms of FL with openset label noise, directly applying centralized solutions locally cannot work and FedPeer significantly improves the performance.

## 2 RELATED WORKS

Federated learning is a collaborative training method to make full use of data from every client without sharing the data. FedSGD (Shokri & Shmatikov, 2015) is the way of FL to pass the gradient between the server and the clients. To improve the performance, FedAvg (McMahan et al., 2017) is proposed and the model weight is passed between the server and the clients. In practice, openset problem is common in FL because the source of every client may vary a lot and it is very likely to find that some of the classes are unique in the specific clients. There are a lot of works to analyze and solve the non-IID problem in FL (Zhao et al., 2018; Li et al., 2019; 2021; Zhang et al., 2021a; Li et al., 2020b; Karimireddy et al., 2020; Andreux et al., 2020).

Label noise is common in the real world (Agarwal et al., 2016; Xiao et al., 2015; Zhang et al., 2017; Wei et al., 2022b). Traditional works on noisy labels usually assume the label noise is *class-dependent*, where the noise transition probability from a clean class to a noisy class *only* depends on the label class. There are many statistically guaranteed solutions based on this assumption (Natarajan et al., 2013; Menon et al., 2015; Liu & Tao, 2015; Liu & Guo, 2020). However, this assumption fails to model the situation where different group of data has different noise patterns (Wang et al., 2021). For example, different clients are likely to have different noisy label spaces, resulting totally different underlying noise transitions. Existing works on federated learning with noisy labels mainly assume the noisy label spaces are identical across different clients (Yang et al., 2022; Xu et al., 2022). There are other notable centralized solutions relying on the memorization effect of a large model (e.g., deep neural network) (Li et al., 2020a; Liu, 2021; Song et al., 2019; Xia et al., 2021; Liu et al., 2020; Cheng et al., 2020). However, in a federated learning system, simply relying on the memorization effect would fail, i.e., the model can perfectly memorize all local noisy samples during local training, since the local data is likely to be imbalanced and with a limited amount (Han et al., 2020; Liu, 2021).

The idea of contrastive labels is to punish the overfitting, which is supposed to avoid memorizing openset local noisy samples. Besides, the concept "openset" is also used in Tuor et al. (2021), where the focus is on the out-of-distribution features and their labels are called openset noise. It is different from ours since they did not focus on in-distribution mislabeled data.

## 3 FORMULATIONS AND MOTIVATIONS

We first formulate the federated learning problem with noisy labels (Section 3.1) then formally define the openset label noise in FL and motivate the necessity of our approach in this setting by showing examples of failures in existing methods (Section 3.2).

### 3.1 FEDERATED LEARNING WITH NOISY LABELS

**Federated learning**   Consider a $K$ class classification problem in a federated learning system with $C$ clients. Each client $c \in [C] := \{1, \cdots, C\}$ holds a local dataset $D_c := \{(x_n^c, y_n^c)\}_{n \in [N_c]}$, where $N_c$ is the number of instances in $D_c$ and $N_c := \{1, \cdots, N_c\}$. Assume there is no overlap among $D_c, \forall c$. Denote the union of all the local datasets by $D := \{(x_n, y_n)\}_{n \in [N]}$. Clearly, we have $D = \cup_{c \in [C]} D_c$ and $N = \sum_{c \in [C]} N_c$. Denote by $\mathcal{D}_c$ the local data distribution, $(X^c, Y^c) \sim \mathcal{D}_c$ the local random variables of feature and label, $\mathcal{D}$ the global/centralized data distribution, and $(X, Y) \sim \mathcal{D}$ the corresponding global random variables. Denote by $\mathcal{X}, \mathcal{X}_c, \mathcal{Y}$, and $\mathcal{Y}_c$ the space of $X, X_c, Y$, and $Y_c$, respectively. FL builds on the following distributed optimization problem: $\min_\theta \sum_{c \in [C]} \frac{N_c}{N} \cdot L_c(\theta)$, where $\boldsymbol{f}$ is the classifier, $\theta$ is the parameter of $\boldsymbol{f}$. To this end, the local training and global model average are executed iteratively. In local training, each client learns a model $\boldsymbol{f}_c : \mathcal{X} \to \mathcal{Y}$ with its local dataset $D_c$ by minimizing the empirical loss $L_c(\theta_c)$ defined as: $L_c(\theta_c) := \frac{1}{N_c} \sum_{n \in [N_c]} \ell(\boldsymbol{f}_c(x_n^c; \theta_c), y_n^c)$, where for classification problems, the loss function is usually the cross-entropy (CE) loss: $\ell(\boldsymbol{f}(X; \theta), Y) = -\ln(f_{(X;\theta)}[Y]), Y \in [K]$. In the following global model average, each client $c$ sends its model parameter $\theta_c$ to the central server, which is further aggregated following FedAvg (McMahan et al., 2017):

$$\theta = \sum_{c \in [C]} \frac{N_c}{N} \cdot \theta_c. \tag{1}$$

### 3.2 OPENSET NOISE IN FEDERATED LEARNING

When the label $y$ is corrupted, the clean dataset $D$ becomes the noisy dataset $\tilde{D} := \{(x_n, \tilde{y}_n\}_{n \in [N]}$ where $\tilde{y}_n$ is the noisy label and possibly different from $y_n$. The noisy data $(x_n, \tilde{y}_n)$ can be viewed as the specific point of the random variables $(X, \tilde{Y})$ which is from the distribution $\tilde{\mathcal{D}}$. Noise transition matrix $T$ characterizes the relationship between $(X, Y)$ and $(X, \tilde{Y})$. The shape of $T$ is $K \times K$ where $K$ is the number of classes in $\mathcal{D}$. The $(i, j)$-th element of $T$ represents the probability of flipping a clean label $Y = i$ to noisy label $Y = j$, i.e., $T_{ij} := \mathbb{P}(\tilde{Y} = j | Y = i)$. If $\tilde{Y} = Y$ always holds, $T$ is an identity matrix. Note the above definition builds on the assumption that $T$ is class-dependent, which is a common assumption in centralized learning with noisy labels (Natarajan et al., 2013; Menon et al., 2015; Liu & Tao, 2015). However, in FL, $T$ is likely to be different for different clients (a.k.a. group-dependent (Wang et al., 2021)). Specifically, we use $T$ to denote the *global* noise transition matrix for $\tilde{D}$ and $T_c$ to denote the *local* noise transition matrix for $\tilde{\mathcal{D}}_c$. In a practical federated learning scenario where the data across different clients are non-IID, different clients may have different label spaces. When the labels are noisy, we naturally have the following definition of openset label noise in FL.

**Definition 1** (Openset noisy labels in FL). *The label noise in client $c$ is called openset if $\tilde{\mathcal{Y}}_c \neq \tilde{\mathcal{Y}}$.*

**Generation of openset noise**   We propose the following noise generation process to model openset label noise in practical FL systems. Denote by $\mathbb{1}_{c,k}$ the indicator random variable that label class $k$ is included in client $c$, where $\mathbb{1}_{c,k} = 1$ (w.p. $Q_{c,k}$) indicates client $c$ has data belonging to class $k$ and $\mathbb{1}_{c,k} = 0$ otherwise. The indicators $\{\mathbb{1}_{c,k} \mid \forall c \in [C], k \in [K]\}$ are generated independently with the probability matrix $Q$, where the $(c, k)$-th element is $Q_{c,k} := \mathbb{E}[\mathbb{1}_{c,k}]$. In practice, if all the elements in $\{\mathbb{1}_{c,k} | k \in [K]\}$ are identical, meaning the client $c$ can observe nothing or all the

classes, then $\{\mathbb{1}_{c,k} | k \in [K]\}$ will be re-generated until client $c$ is an openset client. Denote by $I_k := \{c | \mathbb{1}_{c,k} = 1, c \in [C]\}$ the set of clients that include class $k$. Denote by $\tilde{D}^{(k)} = \{n | \tilde{y}_n = k\}$ the indices of instances that are labeled as class $k$. For each $k \in [K]$, instances in $\tilde{D}^{(k)}$ will be distributed to clients with $\mathbb{1}_{c,k} = 1$ either uniformly or non-uniformly as follows.

- *Uniform allocation:* Randomly sample (without replacement) $|\tilde{D}^{(k)}|/|I_k|$ indices from $\tilde{D}^{(k)}$ and allocate the corresponding instances to client $c$. Repeat for all $c \in I_k$.
- *Non-uniform allocation:* Generate probabilities $\{u_c | c \in I_k\}$ from Dirichlet distribution $\mathsf{Dir}(\mathbf{1})$ with parameter $\mathbf{1} := [1, \cdots, 1]$ ($|I_k|$ values). Randomly sample (without replacement) $|\tilde{D}^{(k)}| \cdot u_c$ indices from $\tilde{D}^{(k)}$ and allocate the corresponding instances to client $c$. Repeat for all $c \in I_k$.

In this way, all the clients have openset label noise, i.e., $\mathcal{Y}_c \neq \tilde{\mathcal{Y}}, \forall c \in [C]$.

**Example** Consider the following example. For a data distribution $(X, Y) \sim \mathcal{D}$ where $Y \in \mathcal{Y} := \{1, 2, \cdots, K\}$, the set of all the opensets is the combination of $\mathcal{Y}$ except the full set of $\mathcal{Y}$ and the empty set. For example, if $\mathcal{Y}$ is $\{1, 2, 3\}$, there would be $2^K - 2 = 6$ different combinations of the noisy label space: $\{1, 2, 3, (1, 2), (1, 3), (2, 3)\}$. It should be noted that it is still possible that the union of all the clients still cannot cover $\mathcal{Y}$. An example of the real and openset $T$ in the 3-class classification problem is as follows. Suppose the *real* noise transition matrix $T_{\text{real}}$ is shown on the LHS. However, if we only observe $\tilde{\mathcal{Y}}_c = \{1, 2\}$ in client $c$, the *optimal estimate* of $T$ relying only on $\tilde{D}_c$ could only be $T_{\text{OptEst}}$ even though we know $D_c$. This is because when $\tilde{\mathcal{Y}}_c = \{1, 2\}$, we have $\mathbb{P}(\tilde{Y} = 3) = 0 \Rightarrow \mathbb{P}(\tilde{Y} = 3 | Y = 3) = 0$, resulting that the other two probabilities have to be normalized from $(1/16, 3/16)$ to $(1/4, 3/4)$ to get a total probability of 1.

$$T_{\text{real}} = \begin{bmatrix} 1 & 0 & 0 \\ 1/3 & 2/3 & 0 \\ 1/16 & 3/16 & 3/4 \end{bmatrix} \qquad T_{\text{OptEst}} = \begin{bmatrix} 1 & 0 & 0 \\ 1/3 & 2/3 & 0 \\ 1/4 & 3/4 & 0 \end{bmatrix}$$

**Openset noise is challenging** A good number of correction approaches in the learning with noisy labels literature would require using the transition matrix $T$. For instance, loss correction (Patrini et al., 2017) is a popular tool to solve the noisy label problem as

$$\ell^{\rightarrow}(\boldsymbol{f}(X), \tilde{Y}) := \ell(T^{\top}\boldsymbol{f}(X), \tilde{Y}) \tag{2}$$

where $T^{\top}$ is the transpose of $T$. The key step of the loss correction approach is to estimate a correct $T$. However, if the label space is openset, the best estimated $T$ will lead to a wrong prediction result. Based on the example above, the best-corrected output is

$$T^{\top}\boldsymbol{f}(X) = \begin{bmatrix} 1 & 1/3 & 1/4 \\ 0 & 2/3 & 3/4 \\ 0 & 0 & 0 \end{bmatrix} \begin{bmatrix} f_1(X; \theta) \\ f_2(X; \theta) \\ f_3(X; \theta) \end{bmatrix} = \begin{bmatrix} f_1(X; \theta) + f_2(X; \theta)/3 + f_3(X; \theta)/4 \\ 2f_2(X; \theta)/3 + 3f_3(X; \theta)/4 \\ 0 \end{bmatrix}, \tag{3}$$

where $\boldsymbol{f} = [f_1, f_2, f_3]^{\top}$. The model cannot distinguish class 3 which is reasonable. However, it will misclassify class 2 to class 3 because class 3 has a larger weight. For example, given an instance $(x, y = 2)$, the cross entropy loss is $-\ln(2f_2(x; \theta)/3 + 3f_3(x; \theta)/4)$ where $f_3(x; \theta) = 1$ leads to the minimization of the loss, making the loss correction fail.

### 3.3 OUR MOTIVATION AND BUILDING IDEA

The above example highlights the challenge of adapting approaches that use noise transition matrix $T$ to our openset FL setting. Therefore, we hope to circle around by building our solutions upon ideas that do not require the knowledge of $T$. Our design is inspired by the idea of the "contrastive labels". Traditional training relies on positive labels, where the loss is denoted by $\ell(f(x_n), \tilde{y}_n)$. However, this kind of loss function is prone to label noise. With a powerful model such as deep neural networks, the noisy distribution will be memorized, inducing generalization error (Liu, 2021).

A number of recent works on noisy labels build on the idea of the contrastive label either explicitly (Wei et al., 2022a; Liu & Guo, 2020) or implicitly (Patrini et al., 2017). For example, a "contrastive label" (Wei et al., 2022a) is defined by a negative label $-\tilde{y}_n$ that introduce a negative loss $-\ell(f(x_n), \tilde{y}_n)$ that punishes the classifier from memorizing the noisy label distribution. Intuitively, if we know the $n$-th label is corrupted, we prefer to make it a negative label rather than the traditional

one since we want to prevent the model from memorizing the wrong pattern. However, in practice, it is challenging to know whether each individual label is corrupted or not. One tractable solution is to use some "random" negative labels. Notable solutions are random peer samples in peer loss (Liu & Guo, 2020). We adopt peer loss as our building block, which is free of the knowledge of noise rates.

To be more concrete, in Liu & Guo (2020), for each example $(x_n, \tilde{y}_n)$, peer loss defines as (an equivalent form):

$$\ell_{\text{PL}}(\boldsymbol{f}(x_n), \tilde{y}_n) := \ell(\boldsymbol{f}(x_n), \tilde{y}_n) - \ell(\boldsymbol{f}(x_n), \tilde{y}_{n'}), \tag{4}$$

where $\tilde{y}_{n'}$ is a randomly sampled peer label. Later as a follow-up work (Cheng et al., 2020), $\ell_{\text{CORES}}$ was proposed as a more stable version of $\ell_{\text{PL}}$ which has the same expectation as $\ell_{\text{PL}}$:

$$\ell_{\text{CORES}}(\boldsymbol{f}(x_n), \tilde{y}_n) = \ell(\boldsymbol{f}(x_n), \tilde{y}_n) - \mathbb{E}_{\mathcal{D}_{\tilde{Y}|\tilde{D}}}[\ell(\boldsymbol{f}(x_n), \tilde{Y}], \tag{5}$$

where $\mathcal{D}_{\tilde{Y}|\tilde{D}}$ is the distribution of $\tilde{Y}$ given dataset $\tilde{D}$. Peer loss and $\ell_{\text{CORES}}$ have strong consistency guarantees. Consider a binary classification problem and let $e_1 := \mathbb{P}(\tilde{Y} = 2|Y = 1)$ and $e_2 = \mathbb{P}(\tilde{Y} = 1|Y = 2)$. Then it was proved in Liu & Guo (2020) the following robustness of peer loss:

**Proposition 1** (Robustness of peer loss (Liu & Guo, 2020)). *Peer loss is invariant to label noise:*

$$\mathbb{E}_{\tilde{D}}[\ell_{PL}(f(X), \tilde{Y})] = (1 - e_1 - e_2) \cdot \mathbb{E}_{\mathcal{D}}[\ell_{PL}(f(X), Y)].$$

*Moreover, when $\mathbb{P}(Y = 1) = 0.5$ and $\ell$ is the 0-1 loss, minimizing peer loss on noisy distribution $\tilde{\mathcal{D}}$ is equivalent to minimizing 0-1 loss on clean distribution $\mathcal{D}$.*

Can we then follow the above idea and implement either $\ell_{\text{PL}}$ or $\ell_{\text{CORES}}$ by requiring each client to sample the "peer label" $\tilde{y}_{n_2}$ locally? Unfortunately, the answer is no. Although these methods can skip estimating $T$, they still need to observe the full data point or full label space to get the correct results - a local sampling for the peer label $y_{n_2}$ will lead to a distribution that does not capture the global one on $\mathbb{P}(\tilde{Y})$, again challenging the theoretical guarantees of the existing results. Furthermore, from Liu & Guo (2020), the success of peer loss and other loss correction approaches, would often reply on a informative label assumption that $e_1 + e_2 < 1$. Due to this requirement, we now show that for openset noise in FL, $y_{n_2}$ cannot be sampled from the local $\tilde{Y}_c$ directly: if only class 1 is observed from $Y = \{1, 2\}$, $e_1 = 0$ and $e_2 = 1$ by definition. Therefore, each client will need the label information shared from the other clients. Our idea is to rebuild the distribution of noisy label $\tilde{Y}$ in the server and send it to the clients so that the clients can be trained with the noisy label $\tilde{Y}$ and $\tilde{Y}$'s distribution.

# 4 PROPOSED METHOD

Recall our design goal is to implement either loss $\ell_{\text{PL}}(\boldsymbol{f}(x_n), \tilde{y}_n)$ or $\ell_{\text{CORES}}(\boldsymbol{f}(x_n), \tilde{y}_n)$ as if data are centralized. If this goal is achieved, our solution in the openset FL setting will inherit the established properties in peer loss. As discussed earlier, the first challenge we have is each local client does not have the information to draw the peer sample label $\tilde{Y} \sim \mathcal{D}_{\tilde{Y}|\tilde{D}}$.

We propose the following label communication-aided algorithm FedPeer, which we also illustrate in Figure 1. There are *two critical stages* to guarantee the success of the proposed methods with good DP protection:

- **Stage 1:** Privacy-preserving global label communication given in Section 4.1
- **Stage 2:** Peer gradient updates at the local client using $\ell_{\text{PL}}$ given in Section 4.2 and the shared label information from Stage 1.

## 4.1 LABEL COMMUNICATION

Label privacy protection is an essential feature of FL so we cannot pass $\tilde{Y}$ to the other clients, directly. To protect privacy, we adopt the label differential privacy (DP) as Definition 2.

**Definition 2** (Label Differential Privacy (Ghazi et al., 2021)). *Let $\epsilon > 0$. A randomized algorithm $\mathcal{A}$ is said to be $\epsilon$-label differentially private ($\epsilon$-labelDP) if for any two training datasets $D$ and $D'$ that differ in the label of a single example, and for any subset $S$ of outputs of $\mathcal{A}$,*

$$\mathbb{P}(\mathcal{A}(D) \in S) \leq e^{\epsilon} \cdot \mathbb{P}(\mathcal{A}(D') \in S).$$

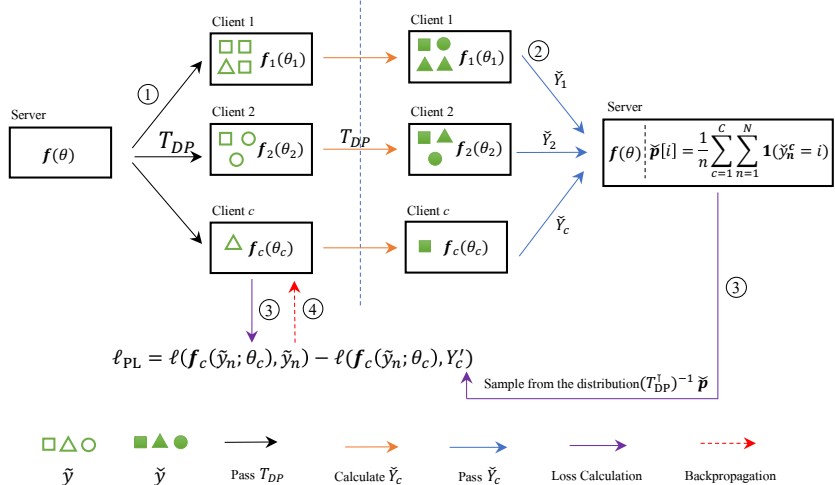

Figure 1: The illustration of FedPeer. *Step 1* is the $T_{\text{DP}}$ generation where the server generates $T_{\text{DP}}$ according to $\epsilon$ and sends it to each client. After receiving $T_{\text{DP}}$, *Step 2* is the label communication. Every client $c$ calculates DP label $\check{Y}_c$ according to $T_{\text{DP}}$ and the noisy label $\tilde{Y}_c$. Clients send $\check{Y}_c$ to the server. The server aggregates every $\check{Y}_c$, calculates the posterior label distribution $\check{p}$ and sends $(T_{\text{DP}}^\top)^{-1}\check{p}$ to every client for the peer term sampling. *Step 3* is the loss calculation using the noisy label $\tilde{Y}_c$ on every client $c$, the model prediction $\hat{Y}_c$ and $Y_c'$ sampled from $(T_{\text{DP}}^\top)^{-1}\check{p}$ and calculate loss. *Step 4* is the back-propagation for peer gradient updates.

The high-level idea is to achieve label privacy (DP), each client $c$ will use a symmetric noise transition matrix $T_{\text{DP}}$ to flip their local labels to protect their labelDP:

$$T_{\text{DP}}[y, \tilde{y}] := \mathbb{P}(\widetilde{Y} = \tilde{y} | Y = y) = \begin{cases} \frac{e^\epsilon}{e^\epsilon + K - 1}, & \text{if } \tilde{y} = y, \\ \frac{1}{e^\epsilon + K - 1}, & \text{if } \tilde{y} \neq y. \end{cases}$$

where $K$ is the number of classes. Then only the flipped labels are shared between the clients and the server. It is easy to show that sharing the flipped labels using $T_{\text{DP}}$ suffices to preserve labelDP:

**Theorem 1** (Label Privacy in FedPeer). *Label sharing in FedPeer is $\epsilon$-labelDP.*

Since we have already known that $\ell_{\text{CORES}}$ is equal to $\ell_{\text{PL}}$ in expectation so we can calculate the second term of $\ell_{\text{PL}}$ only with the distribution $\mathbb{P}(\tilde{Y}|\tilde{D})$. Denote by $\tilde{p}_n^c$ the one-hot encoding of $\tilde{y}_n^c$. The whole label communication process is presented in Algorithm 1. At the beginning of the algorithm, the server will initialize $T_{\text{DP}}$ according to $\epsilon$ and broadcast $T_{\text{DP}}$ to all $C$ clients. For each client $c$, it calculates the DP label distribution of every data point $(x_n^c, \tilde{y}_n^c)$ as $\check{p}_n^c = T_{\text{DP}}^\top \tilde{p}_n^c$, where $\tilde{p}_n^c$ is the distribution of DP label in client $c$. With this distribution, the client generates the DP private label $\check{y}_n^c, n \in [N_c]$ for every data point and every client sends all $\check{y}_n^c$ back to the server. After obtaining all $\check{y}_n^c$ from the clients, the server aggregates the label and calculates the posterior label distribution $\check{p}$. To restore the correct distribution of $\tilde{Y}$, the server calculates $(T_{\text{DP}}^\top)^{-1}\check{p}$. Note that $(T_{DP}^\top)^{-1}T_{DP}^\top(\sum_{i=1}^C \tilde{p}_n^c)/C = \tilde{p}$. To apply $T_{\text{DP}}$ and $(T_{\text{DP}})^{-1}$ sequentially, FedPeer enables the clients to share the information with the others while DP is guaranteed. Finally, the client calculates the local loss according to Equation 4 where $\tilde{Y}$ sampled from $\mathbb{P}(\tilde{Y} = i) := ((T_{\text{DP}}^\top)^{-1}\check{p})[i]$. This label communication procedures guarantees $\epsilon$-DP.

## 4.2 FEDPEER

Based on the distribution $\mathbb{P}(\tilde{Y}|\tilde{D})$, we propose a novel framework based on FedAvg, *FedPeer* to solve openset noise problem. Denote by $\Delta_c^{(r)} := \theta_c^{r+1} - \theta_c^r$, the variation of model parameters in the $r$-th round of the local training in client $c$. Recall $\theta_c$ is the parameter of $\boldsymbol{f}_c$.

As shown in Proposition 1, in centralized training, $\ell_{\text{PL}}$ is proved to be robust to label noise. For each local client, if the label of the second term of $\ell_{\text{PL}}$ is sampled from the distribution of $\mathbb{P}(\tilde{Y}|\tilde{D})$,

---

**Algorithm 1** Label Communication in FedPeer

---

1: **Initialization:** The server initialize $T_{\mathrm{DP}}$ according to $\epsilon$ and broadcast $T_{\mathrm{DP}}$ to all clients.
2: **for** $c$ in $C$ clients **do** ▷ Client label differential privacy protection
3:     calculate $\check{\boldsymbol{p}}_n^c = T_{\mathrm{DP}}^\top \tilde{\boldsymbol{p}}_n^c, \forall n \in [N_c]$.
4:     generate the private label $\check{y}_n^c$ using $\mathbb{P}(\check{y}_n^c = i) = \check{\boldsymbol{p}}_n^c[i], \forall i \in [K], n \in [N_c]$.
5:     send $\{\check{y}_n^c\}_{n \in [N_c]}$ to the server
6: **end for**
7: The server aggregates the label $\{\check{y}_n^c\}_{n \in [N_c]}$ sent from all $C$ clients.
8: The server calculates the posterior label distribution $\check{\boldsymbol{p}}$: $\check{\boldsymbol{p}}[i] := \frac{1}{N} \sum_{c=1}^C \sum_{n=1}^N \mathbb{1}(\check{y}_n^c = i)$.
9: The server calculates $(T_{\mathrm{DP}}^\top)^{-1} \check{\boldsymbol{p}}$ and sends it to each client $c$.
10: The client $c$ samples the $\tilde{Y}$ in Eqn. (4) following $\mathbb{P}(\tilde{Y} = i) = ((T_{\mathrm{DP}}^\top)^{-1} \check{\boldsymbol{p}})[i]$.

---

there would be no difference between FedPeer and centralized training in convergence although the client is openset noisy. Denote by $\Delta^{(r)} := \theta^{r+1} - \theta^r$ the variation of model parameters in the $r$-th round of the corresponding global gradient descent update assuming the local data are collected to a central server. Define $\mathbb{P}(\mathcal{D}_c | \mathcal{D}) := \mathbb{P}((X, Y) \sim \mathcal{D}_c \mid (X, Y) \sim \mathcal{D})$. In detail, it is calculated as $N_c/N$ for client $c$ given $D$. We have the following theorem that guarantees the calibration property of FedPeer.

**Theorem 2** (Local clients with FedAvg). *The aggregated model update of FedPeer is the same as the corresponding centralized model update, i.e.,*

$$\sum_{c \in [C]} \mathbb{P}(\mathcal{D}_c | \mathcal{D}) \cdot \Delta_c^{(r)} = \Delta^{(r)},$$

The details of FedPeer are shown in Algorithm 2. At the beginning of FedPeer, the server and the clients $c$ initialize the model and each client $c$ initialize its own dataset $D_c = \{X_c, \tilde{Y}_c\}$ and loss function $\ell$. After this, the server generates the DP matrix $T_{\mathrm{DP}}$ and sends it to every client and every client $c$ can generate DP labels $\check{y}_n^c$. Next, every client $c$ sends $\check{y}_n^c$ to the server, and the server aggregates DP labels according to Section 4.1. After aggregation at the server, a posterior label distribution $\check{\boldsymbol{p}}$ can be computed and the server sends $(T_{\mathrm{DP}}^\top)^{-1} \check{\boldsymbol{p}}$ back to the client so that the client can sample the peer label from this distribution.

Until now, the preparation is finished and we can start the training process. To simulate the practical usage, only part of rather than all clients will participate in the training in one round. The clients are chosen randomly according to the federated fraction $\lambda$. The selected clients sample the "contrastive label" $Y_c'$ from the distribution $(T_{\mathrm{DP}}^\top)^{-1} \check{\boldsymbol{p}}$ and calculate the loss $L_c$ according to (Liu & Guo, 2020) by using the output of the model $\hat{Y}_c$. The model weight is updated by $L_c$ and the server weight is averaged according to FedAvg (McMahan et al., 2017), which is the end of one communication round.

## 5 EXPERIMENTS AND RESULTS

### 5.1 EXPERIMENTS SETUP

To validate the generality and effectiveness of FedPeer, we select several public datasets with various levels of difficulties including CIFAR-10, CIFAR-100 (Krizhevsky et al., 2009) as benchmark datasets and Clothing-1M (Xiao et al., 2015), CIFAR-N (Wei et al., 2022b) as real-world datasets. To simulate the practical usage, we first apply the noise on the label and generate the openset candidates according to the number of classes $K$ for every client because only the noisy label is visible to the client in the real world. On CIFAR-10 and CIFAR-100, we apply the symmetric noise for benchmark testing while we apply random noise for practical simulation. Furthermore, we also test the performance using Clothing-1M and CIFAR-N to test the performance of FedPeer in real-world scenarios.

For baseline methods, we use FedAvg (McMahan et al., 2017), forward loss correction (LC) (Patrini et al., 2017), FedProx (Li et al., 2020b), Co-teaching (Han et al., 2018) and T-revision (Xia et al.,

---

**Algorithm 2** FedPeer.

---

1: **Server:** initialize model $\boldsymbol{f}_g$, global step size $\alpha_g$ and global communication round $R$.
2: **Each Client** $c$**:** initialize model $\boldsymbol{f}_c$ , the dataset $D_c = \{(x_n^c, \tilde{y}_n^c)\}_{n \in [N_c]}$, local learning rate $\alpha_c$ and local updating iterations $E$.
3: The server generates and broadcasts $T_{\mathrm{DP}}$ to all clients according to Definition 2.
4: Clients generate DP labels $\check{y}_n^c$ and send $\check{y}_n^c$ to the server according to Section 4.1.
5: The server aggregates $\check{y}_n^c$ and calculate the posterior label distribution $\check{\boldsymbol{p}}$.
6: The server send $(T_{\mathrm{DP}}^\top)^{-1}\check{\boldsymbol{p}}$ to each client.
7: **for** $i = 1 \to R$ **do**
8:     Randomly select $C'$ clients from $C$ according to $\lambda$
9:     **for** $c$ in $C'$ clients **do**
10:         $\boldsymbol{f}_c \leftarrow \boldsymbol{f}$                                  ▷ Receive the updated weight from the server
11:         **for** $j = 1 \to E$ **do**
12:             $\hat{y}_n^c \leftarrow \boldsymbol{f}_c(x_n^c), \forall n \in [N_c]$                       ▷ Model Prediction
13:             Sample $(y_c^n)'$ according to $(T_{\mathrm{DP}}^\top)^{-1}\check{\boldsymbol{p}}, \forall n \in [N_c]$.
14:             Calculate loss as $L_c \leftarrow \frac{1}{N_c} \sum_{n=1}^{N_c} (\ell(\hat{y}_n^c, \tilde{y}_n^c) - \ell(\hat{y}_n^c, (y_c^n)'))$      ▷ Calculate $\ell_{\mathrm{PL}}$
15:             $\boldsymbol{f}_c \leftarrow \boldsymbol{f}_c - \alpha_c \cdot \nabla L_c$                      ▷ Model weight updated
16:         **end for**
17:     **end for**
18:     $\boldsymbol{f} \leftarrow \boldsymbol{f} - \alpha_g \cdot \sum_{c=1}^{C'} (\boldsymbol{f}_c - \boldsymbol{f})$                            ▷ FL aggregation
19: **end for**

---

2019) methods. The local updating iteration $E$ is 5 and the federated fraction $\lambda$ is 0.1. The architecture of the network is ResNet-18 (He et al., 2016) for CIFAR dataset and ResNet-50 (He et al., 2016) with ImageNet (Deng et al., 2009) pre-trained weight for Clothing-1M. The local learning rate $\alpha_l$ is 0.01 and the batch size is 32. The total communication round with the server $R$ is 300 and differential privacy $\epsilon$ are 3.58, 5.98 and 3.95 for CIFAR-10, CIFAR-100 and Clothing-1M, respectively. All the experiments are run for 3 times with different random seeds to validate the generality of our methods. Due to the privacy issue in federated learning settings, it is hard to find the best model for practical usage. Thus, we report both the best in the following tables and the last accuracy in Appendix on the testing set. The details of the implementation of every baseline method in the FL setting can be found in the Appendix.

## 5.2 Synthetic Label Noise

There are two strategies to synthesize the openset label noise in FL.

- Symmetric: We first add symmetric label noise (Xia et al., 2019; Han et al., 2018) to dataset $D$ and get $\tilde{D}$, then distribute $\tilde{D}$ to $\tilde{D}_c, \forall c$ following the uniform allocation in Section 3.2. The transition matrix $T$ for the symmetric label noise satisfies $T_{ij} = \eta/(K-1), \forall i \neq j$ and $T_{ii} = 1 - \eta, \forall i \in [K]$, where $\eta \in \{0.2, 0.4, 0.6, 0.8\}$ is the average noise rate.

- Random: We first add random label noise (Zhu et al., 2022) to dataset $D$ and get $\tilde{D}$, then distribute $\tilde{D}$ to $\tilde{D}_c, \forall c$ following the non-uniform allocation in Section 3.2. The $T$ of random noise is generated as follows. The diagonal elements of $T$ for the random label noise is generated by $\eta + \mathsf{Unif}(-0.05, 0.05)$, where $\eta$ is the average noise rate, $\mathsf{Unif}(-0.05, 0.05)$ is the uniform distribution bounded by $-0.05$ and $0.05$. The off-diagonal elements in each row of $T$ follow the Dirichlet distribution $(1 - T_{ii}) \cdot \mathsf{Dir}(\mathbf{1})$, where $\mathbf{1} = [1, \cdots, 1]$ ($K-1$ values). The random strategy is more practical than the symmetric one.

**Results and Discussion**    Table 1 shows FedPeer is significantly better than all the baseline methods in the symmetric strategy across almost all the noise rate settings. It is also better than the other methods in most settings of the random strategy. FedPeer is very competitive in all the settings. Table 1 also shows directly applying the methods for centralized learning with noisy labels cannot be statistically better than the traditional federated learning solution (FedAvg) and its adapted version (FedProx), indicating the openset label noise in FL is indeed challenging and special treatments are necessary to generalize the centralized solution to the FL setting. We also report the accuracy of the last epoch in Table 5 and 6 in Appendix.

Table 1: The performance (the best accuracy) of all methods on CIFAR-10 and CIFAR-100

| Dataset | Methods | Symmetric | | | | Random | | | |
|---|---|---|---|---|---|---|---|---|---|
| | | 0.2 | 0.4 | 0.6 | 0.8 | 0.2 | 0.4 | 0.6 | 0.8 |
| CIFAR-10 | FedAvg | 76.84±0.91 | 63.34±1.82 | 43.83±0.51 | 22.13±1.25 | 76.24±1.58 | 59.19±1.01 | 46.80±2.63 | 21.80±0.28 |
| | LC | 79.14±0.35 | 63.57±0.61 | 44.33±1.13 | 22.98±1.60 | 74.96±1.92 | 61.49±3.02 | 40.52±2.18 | 23.84±3.37 |
| | FedProx | 67.31±0.38 | 57.76±0.72 | 43.84±0.78 | **25.08±0.35** | 65.42±0.62 | 54.51±0.73 | 46.08±1.22 | 26.77±0.42 |
| | Co-teaching | 78.64±0.45 | 70.60±0.47 | 48.63±0.57 | 21.06±2.10 | 75.11±0.39 | 59.00±1.19 | 31.30±2.03 | 17.10±3.78 |
| | T-revision | 69.16±6.20 | 51.86±6.64 | 31.93±2.56 | 15.27±1.87 | 64.69±5.08 | 46.22±1.17 | 31.81±2.83 | 17.12±0.73 |
| | FedPeer | **84.77±0.12** | **75.75±1.96** | **55.50±1.33** | 24.64±0.55 | **82.15±0.24** | **72.69±1.57** | **59.06±1.38** | **27.55±1.49** |
| CIFAR-100 | FedAvg | 47.78±0.50 | 32.63±0.27 | 20.32±0.51 | 10.62±0.26 | 47.75±0.29 | 31.06±0.79 | 20.14±0.32 | 9.71±0.43 |
| | LC | 48.92±0.42 | 33.15±0.23 | 20.39±0.36 | 10.43±0.45 | 49.03±0.17 | 32.67±0.75 | 19.78±0.67 | 10.13±0.36 |
| | FedProx | 40.42±4.04 | 29.21±0.49 | 20.36±0.68 | **12.60±0.65** | 37.38±0.38 | 28.44±0.25 | **21.38±0.03** | **11.64±0.64** |
| | Co-teaching | 41.15±0.28 | 29.81±0.72 | 18.01±0.28 | 8.73±1.08 | 40.55±1.79 | 28.51±1.41 | 18.47±1.95 | 6.56±1.38 |
| | T-revision | 48.21±0.56 | 31.35±0.46 | 17.41±0.22 | 7.79±0.28 | 48.24±0.47 | 30.91±0.55 | 16.95±0.78 | 7.46±0.20 |
| | FedPeer | **53.39±0.43** | **34.99±1.66** | **21.35±0.69** | 11.02±0.66 | **51.73±0.36** | **34.43±0.72** | 21.35±0.72 | 10.64±0.43 |

Table 2: The performance (the best accuracy) of all methods on CIFAR-N and Clothing-1M

| Datasets | CIFAR-10 | | | CIFAR-100 | Clothing-1M |
|---|---|---|---|---|---|
| Methods | Worst | Random | Aggregate | Fine | 1M Noisy Training |
| FedAvg | 46.55±7.82 | 59.69±4.88 | 66.41±6.52 | 22.65±2.29 | 70.27 |
| LC | 46.67±8.21 | 59.27±5.72 | 67.27±4.76 | 22.59±1.66 | 70.05 |
| FedProx | 58.47±0.97 | 69.35±0.62 | 74.48±1.00 | 35.33±0.35 | 65.96 |
| Co-teaching | 24.80±2.27 | 47.34±21.05 | 62.04±11.26 | 17.83±0.39 | 40.33 |
| T-revision | 57.85±19.44 | 55.06±8.40 | 63.40±9.99 | 22.18±1.44 | 66.95 |
| FedPeer | **58.51±10.89** | **73.68±4.35** | **76.51±9.30** | **40.60±1.91** | **70.88** |

Table 3: The comparison of the influence of different $\epsilon$ on the performance.

| $\epsilon = 1$ | $\epsilon = 2$ | $\epsilon = 4$ | $\epsilon = 8$ | $\epsilon = 100$ | $\epsilon = 3.58$ |
|---|---|---|---|---|---|
| 72.47±2.64 | 71.60±1.96 | 72.27±1.87 | 73.00±1.96 | 73.75±2.38 | 72.44±1.52 |

## 5.3 REAL-WORLD LABEL NOISE

We also test the performance on two real-world datasets: CIFAR-N (Wei et al., 2022b) and Clothing-1M (Xiao et al., 2015). Different from the benchmark datasets, these datasets are corrupted naturally. Clothing-1M is collected from the real website where both data and labels are from the real users. The noisy ratio is about 0.4 in Clothing-1M. CIFAR-N consists of CIFAR-10 and CIFAR-100. $\tilde{D}_c$ is generated according to the random setting given in Section 5.2. The labels of CIFAR-N are collected from the human annotation. There are three levels of noisy ratio in CIFAR-10, *worst*, *aggregate* and *random* while there is only one noisy level in CIFAR-100. It can be found that FedPeer outperforms all the baseline methods in the real-world dataset, showing great potential in practical usage.

## 5.4 EFFECT OF DP LEVEL

According to Section 4.1 and 4.2, label communication and peer gradient updates at local clients are two key steps in FedPeer. $\epsilon$ is the parameter to control the level of DP protection. Following Ghazi et al. (2021), we study the influence of $\epsilon$ on the performance. We select the CIFAR-10 corrupted by random noise whose ratio is 0.4 as the method. All the experiments are run with 10 random seeds. In terms of the randomness of model initialization and the noise generation, it can be found that FedPeer is stable with the change of $\epsilon$, which agrees with our theoretical guarantee.

## 6 CONCLUSION

We have defined openset label noise in FL and proposed FedPeer to use globally communicated contrastive labels to prevent local models from memorizing openset noise patterns. We have proved that FedPeer is able to approximate a centralized solution with strong theoretical guarantees. Our experiments also verified the advantage of FedPeer. Admittedly, FedPeer is only tested with different label noise regimes with synthetic data partitions. Future works include testing FedPeer with real-world FL data partitions and real-world clients such as mobile devices.

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

APPENDIX

**Roadmap** The appendix is composed as follows. Section A presents all the notations and their meaning we use in this paper. Section C introduces the implementation details of the experiments and how to apply the centralized training methods to FL. Section D shows the experiment results with more details that are not given in the main paper due to the page limit.

## A    NOTATION TABLE

| Notation | Explanation |
|---|---|
| $\eta$ | Noisy ratio |
| $C$ | Total number of clients |
| $c$ | Client $c$ in federated learning |
| $\tilde{Y}$ | Random variables for the noisy label |
| $\hat{Y}$ | Random variables for the output of the model |
| $K$ | Number of classes in $\mathcal{Y}$ |
| $T$ | Transition matrix |
| $\mathbb{P}$ | The probability |
| $\mathbb{E}$ | The expectation |
| $\lambda$ | Federated fraction to control the number of clients in every round |
| $\alpha_g$ | The global step size on the server side |
| $\alpha_l$ | The local learning rate on the client side |
| $L, \ell$ | The loss, the loss function |
| $R$ | The global communication round |
| $E$ | The local updating round |
| $T_{\text{DP}}$ | Differential privacy transition matrix |
| $T$ | Transition matrix |
| $\mathcal{A}$ | The label communication algorithm |
| $\check{Y}_c$ | Labels protected by differential privacy |
| $\tilde{\boldsymbol{p}}_n^c$ | one-hot encoding of $\tilde{y}_n^c$ |
| $\check{\boldsymbol{p}}$ | The posterior label distribution after differential privacy corruption |
| $\theta, \theta_c^r$ | The model parameters, the model parameters of client $c$ at $r$-th round |
| $\Delta_c^{(r)}$ | The variation of model parameters in $r$-th round of the client $c$ |
| $X, Y$ | Random variables for the feature and label |
| $\mathcal{X}, \mathcal{Y}$ | The space of $X, Y$ |
| $\boldsymbol{f}_c, \boldsymbol{f}_g$ | The client model, The global model |
| $N, N_c$ | Total number of samples, number of samples in client $c$ |
| $(x_n^c, y_n^c)$ | The $n$-th example in the client $c$ |
| $D_c := \{(x_n^c, y_n^c)\}_{n \in [N_c]}$ | Dataset of client $c$ |
| $D := \{(x_n, y_n)\}_{n \in [N]}$ | Dataset |
| $I_k := \{c \mid \mathbb{1}_{c,k} = 1, c \in [C]\}$ | The vector indicating whether client $c$ can access class $k$ or not |

Table 4: Table of notations used in the paper

## B    PROOFS

In this section, we present all the proofs of the theorems.

### B.1    PROOF OF THEOREM 1

*Proof.* Denote by $\mathcal{A}$ the label communication algorithm, where the input is $y$ and the output is $y_{\text{DP}}$. Then after flipping the label $y$ according to the noise transition matrix $T$, we have

$$\mathbb{P}(\mathcal{A}(y) = y_{\text{DP}}) = \begin{cases} \frac{e^\epsilon}{e^\epsilon + K - 1}, & \text{if } y_{\text{DP}} = y, \\ \frac{1}{e^\epsilon + K - 1}, & \text{if } y_{\text{DP}} \neq y. \end{cases}$$

Accordingly, for another label $y'$, we have

$$\mathbb{P}(\mathcal{A}(y') = y_{\text{DP}}) = \begin{cases} \frac{e^\epsilon}{e^\epsilon + K - 1}, & \text{if } y_{\text{DP}} = y', \\ \frac{1}{e^\epsilon + K - 1}, & \text{if } y_{\text{DP}} \neq y'. \end{cases}$$

Then the quotient of two probabilities can be upper bounded by

$$\frac{\mathbb{P}(\mathcal{A}(y) = y_{\text{DP}})}{\mathbb{P}(\mathcal{A}(y') = y_{\text{DP}})} \le e^{\epsilon}.$$

With Definition 2, we know the above equation is exactly the definition of $\epsilon$-labelDP, i.e., the label sharing algorithm is $\epsilon$-labelDP.

$\square$

### B.2 PROOF OF THEOREM 2

*Proof.* The centralized peer loss on $\mathcal{D}$ is

$$\mathbb{E}_{\mathcal{D}}[\ell_{\text{peer}}(f(X), \widetilde{Y})] = \mathbb{E}_{\mathcal{D}}\left[\ell(f(X), \widetilde{Y}) - \beta \cdot \mathbb{E}_{\mathcal{D}_{\widetilde{Y}'|\widetilde{D}}}[\ell(f(X), \widetilde{Y}')]\right].$$

For each client $i$, the local FedPeer loss is

$$\mathbb{E}_{\mathcal{D}_i}[\ell_{\text{FedPeer}}(f(X), \widetilde{Y})] = \mathbb{E}_{\mathcal{D}_i}\left[\ell(f(X), \widetilde{Y}) - \beta \cdot \mathbb{E}_{\mathcal{D}_{\widetilde{Y}'|\widetilde{D}}}[\ell(f(X), \widetilde{Y}')]\right].$$

Denote by $\mathbb{P}(\mathcal{D}_i|\mathcal{D})$ the probability of drawing a data point from client $i$. We have

$$\sum_{i \in [C]} \mathbb{P}(\mathcal{D}_i|\mathcal{D}) = 1.$$

Then

$$\sum_{i \in [C]} \mathbb{P}(\mathcal{D}_i|\mathcal{D})\mathbb{E}_{\mathcal{D}_i}[\ell_{\text{FedPeer}}(f(X), \widetilde{Y})]$$

$$= \sum_{i \in [C]} \mathbb{P}(\mathcal{D}_i|\mathcal{D})\mathbb{E}_{\mathcal{D}_i}\left[\ell(f(X), \widetilde{Y}) - \beta \cdot \mathbb{E}_{\mathcal{D}_{\widetilde{Y}'|\widetilde{D}}}[\ell(f(X), \widetilde{Y}')]\right]$$

$$= \mathbb{E}_{\mathcal{D}}\left[\ell(f(X), \widetilde{Y}) - \beta \cdot \mathbb{E}_{\mathcal{D}_{\widetilde{Y}'|\widetilde{D}}}[\ell(f(X), \widetilde{Y}')]\right]$$

$$= \mathbb{E}_{\mathcal{D}}[\ell_{\text{peer}}(f(X), \widetilde{Y})].$$

Each round may include multiple epochs. Suppose there are $t$ local epochs. The variation of model parameters in the $r$-th round of the local training in client $i$ can be decomposed by

$$\begin{aligned}
\Delta_i^{(r)} &:= \theta_i^{(r+1)} - \theta_i^{(r)} \\
&= \theta_i^{(r+1,t)} - \theta_i^{(r+1,t-1)} + \theta_i^{(r+1,t-1)} + \cdots - \theta_i^{(r,1)} \\
&= \frac{\partial \mathbb{E}_{\mathcal{D}_i}[\ell_{\text{FedPeer}}(f(X), \widetilde{Y}; \theta)]}{\partial \theta}\bigg|_{\theta = \theta_i^{(r+1,t-1)}} + \cdots + \frac{\partial \mathbb{E}_{\mathcal{D}_i}[\ell_{\text{FedPeer}}(f(X), \widetilde{Y}; \theta)]}{\partial \theta}\bigg|_{\theta = \theta_i^{(r+1,1)}}.
\end{aligned}$$

Therefore,

$$\begin{aligned}
\sum_{i \in [C]} \mathbb{P}(\mathcal{D}_i|\mathcal{D})\Delta_i^{(r)} &= \frac{\partial \sum_{i \in [C]} \mathbb{P}(\mathcal{D}_i|\mathcal{D})\mathbb{E}_{\mathcal{D}_i}[\ell_{\text{FedPeer}}(f(X), \widetilde{Y}; \theta_i^{(r+1,t-1)})]}{\partial \theta} \\
&\quad + \cdots + \frac{\partial \sum_{i \in [C]} \mathbb{P}(\mathcal{D}_i|\mathcal{D})\mathbb{E}_{\mathcal{D}_i}[\ell_{\text{FedPeer}}(f(X), \widetilde{Y}; \theta_i^{(r+1,1)})]}{\partial \theta} \\
&= \frac{\partial \mathbb{E}_{\mathcal{D}}[\ell_{\text{peer}}(f(X), \widetilde{Y}; \theta^{(r+1,t-1)})]}{\partial \theta} \\
&\quad + \cdots + \frac{\partial \mathbb{E}_{\mathcal{D}}[\ell_{\text{peer}}(f(X), \widetilde{Y}; \theta^{(r+1,1)})]}{\partial \theta} \\
&= \Delta^{(r)}.
\end{aligned}$$

$\square$

## C  IMPLEMENTATION DETAILS

**Platform and Programming Environment**  We train our model on NVIDIA RTX A500 server with torch and torchvision 1.10 and 0.11, respectively. The details of the baseline methods are as follows.

**Loss correction**  We apply FedAvg in the first 150 rounds to make the weight stable. At the 150th round, the transition matrix of every client will be estimated according to the confidential score of 95%. The predicted label whose confidential score is over 95% is considered as the ground truth so that we can get every transition matrix of every client. We apply loss correction in the rest 150 rounds according to Equation 2.

**Co-teaching**  Co-teaching uses two same networks to distinguish the noisy data and the clean data. Similarly, we initialize two same networks when the client initializes and update the two clients in the same way as the original co-teaching network. The server also keeps two models. In every communication round, the weights of the two models will average correspondingly.

**T-revision**  T-revision consists of three steps: estimation of $T$, loss correction, and T-revision. In the first 20 communication rounds, the selected clients update the weight at every communication round and all the clients estimate $T_c$. After the 20th round, the selected clients at every communication apply forward loss correction for another 140 rounds. After the 160th round, we apply T-revision.

**DivideMix**  DivideMix uses two same networks to distinguish the noisy label. One network is used to assign the pseudo label, the other network is used to the classification. The pseudo label is generated by a Gaussian mixture process. In addition, DivideMix uses mix-up data augmentation to boost performance. In FL paradigm, every client will maintain two clients and do the same operation as the centralized training in DivideMix.

## D  EXPERIMENT RESULTS

In addition to the results of the best accuracy of the methods, we also present the accuracy of the last epoch. The results of symmetric noise are given in Table 5. The results of random noise are given in Table 6. The results of real-world cases are given in Table 7.

| Dataset | Methods | 0.2 | 0.4 | 0.6 | 0.8 |
|---------|---------|-----|-----|-----|-----|
| | FedAvg | 66.90±6.26 | 51.87±6.26 | 32.73±4.90 | 15.79±1.51 |
| | LC | 67.49±3.25 | 49.17±9.87 | 32.32±4.57 | 15.28±2.09 |
| | FedProx | 66.57±0.35 | 53.50±0.23 | 38.01±0.66 | 20.28±0.90 |
| CIFAR-10 | Co-teaching | 76.17±1.72 | **69.03±2.25** | **46.66±3.75** | 17.88±2.10 |
| | T-revision | 63.62±3.75 | 49.76±3.99 | 29.83±3.85 | 15.34±1.02 |
| | FedPeer | **78.59±5.12** | 58.35±6.61 | 46.22±1.31 | **20.87±2.37** |
| | FedAvg | 46.79±0.79 | 30.59±0.28 | 17.15±0.58 | 7.53±0.26 |
| | LC | 47.66±0.19 | 30.95±0.30 | 17.27±0.19 | 7.43±0.06 |
| | FedProx | 37.32±0.82 | 28.27±0.40 | **19.80±1.05** | **11.41±0.31** |
| CIFAR-100 | Co-teaching | 40.90±0.11 | 29.41±0.64 | 17.45±0.46 | 8.27±0.96 |
| | T-revision | 48.02±0.05 | 31.22±0.40 | 17.48±0.72 | 7.65±0.28 |
| | FedPeer | **52.58±0.84** | **33.17±1.16** | 18.10±0.32 | 7.78±0.04 |

Table 5: The performance (the last accuracy) of our methods on CIFAR-10 and CIFAR-100. The noise is symmetric. We compare different methods under different noisy ratios.

| Dataset | Methods | 0.2 | 0.4 | 0.6 | 0.8 |
|---------|---------|-----|-----|-----|-----|
| | FedAvg | 59.53±3.39 | 43.14±5.51 | 30.16±3.73 | 16.73±2.11 |
| | LC | 60.63±6.31 | 43.09±5.71 | 30.11±2.28 | 15.15±1.89 |
| | FedProx | 62.86±4.00 | 47.62±5.42 | 36.06±4.14 | **19.16±0.64** |
| CIFAR-10 | Co-teaching | 64.81±3.91 | 54.41±0.80 | 26.02±2.66 | 13.43±2.68 |
| | T-revision | 63.66±5.88 | 39.70±16.35 | 34.17±4.24 | 16.78±0.81 |
| | FedPeer | **68.18±8.79** | **63.74±2.25** | **40.41±1.97** | 16.09±1.71 |
| | FedAvg | 46.36±0.18 | 30.01±1.01 | 16.78±0.37 | 7.21±0.42 |
| | LC | 48.34±0.50 | 30.69±1.19 | 17.11±0.40 | 7.80±0.04 |
| | FedProx | 37.25±0.52 | 27.70±0.05 | **19.92±0.45** | **10.49±0.08** |
| CIFAR-100 | Co-teaching | 36.08±6.17 | 25.57±3.61 | 18.38±1.81 | 6.49±1.45 |
| | T-revision | 48.01±0.83 | 30.93±0.66 | 17.10±0.55 | 7.57±0.63 |
| | FedPeer | **51.08±0.08** | **33.24±1.09** | 17,61±0.27 | 8.16±0.37 |

Table 6: The performance (the last accuracy) of our methods on CIFAR-10 and CIFAR-100. The noise is random. We compare different methods under different noisy ratios.

| Datasets | CIFAR-10 | | | CIFAR-100 | Clothing-1M |
|----------|----------|--|--|-----------|-------------|
| Methods | Worst | Random | Aggregate | Fine | 1M Noisy Training |
| FedAvg | 41.85±5.83 | 57.33±6.45 | 65.00±6.51 | 22.13±2.54 | 66.91 |
| LC | 41.37±5.66 | 58.20±5.58 | 65.44±6.51 | 21.99±1.54 | 67.75 |
| FedProx | 51.10±7.52 | **65.19±5.73** | 72.30±2.11 | 34.77±0.01 | 66.97 |
| Co-teaching | 24.11±2.03 | 44.10±18.21 | 60.28±10.54 | 16.77±0.25 | 40.37 |
| T-revision | **56.16±21.37** | 50.90±8.67 | 58.04±18.13 | 22.52±1.32 | 66.79 |
| FedPeer | 51.95±7.33 | 64.50±12.05 | **74.40±8.91** | **39.78±1.72** | **69.70** |

Table 7: The performance (the last accuracy) of all methods on CIFAR-N and Clothing-1M.

