# OpenReview forum: "Federated Learning with Openset Noisy Labels"
_ICLR.cc/2023/Conference — Submitted to ICLR 2023_

### Official Review · Reviewer_GPjn · 2022-10-24

**Confidence:** 5
**Clarity, Quality, Novelty And Reproducibility:** 1.	Clear writing and presentation, ex…
**Correctness:** 3
**Technical Novelty And Significance:** 2
**Empirical Novelty And Significance:** Not applicable
**Recommendation:** 5

**Strength And Weaknesses:**

Strength:
1. Paper is straightforward and easy to follow, problem is compelling and ideas communicated clearly.

2. Experimental analysis and comparison presented show that the method is useful.

Weaknesses:

1. The openset noise problem defined in this paper is that the observed label space at these clients may be differ as well, even though their underlying clean labels are drawn from the same label space. How is this problem different from the label noise problem in non-iid federated learning scenarios with skewed label distributions?

2. The previous description of the problem is relatively clear, and the example in section 3.2 can be placed in the appendix.

3. Whether the server can use the collected DP label distribution to estimate the original label distribution through the noise transition matrix, which brings the risk of privacy leakage to some extent.

4. When using the symmetric strategy to synthesize openset label noise, the labels are evenly labeled as the wrong class. Does this weaken the difference in label spaces between clients, and thus deviate from the problem scenario addressed by this paper.

5. From the experimental results, the effectiveness of the method is significant when the noise rate is small, but the method seems to fail when the noise rate is high. How to explain this phenomenon?


**Summary Of The Paper:**

The paper defines a openset noise problem in federated learning, and proposes a corresponding solution FedPeer. FedPeer has two critical stages: 1) The author adopts label differential privacy to share the label space of the client, thereby realizing privacy-preserving global label communication. 2) The authors sample the contrastive labels from the global label distribution obtained in the previous stage, and use the output of the local model to compute the peer loss, enabling robust local gradient updates. The proposed method is validated through quantitative comparison with baselines, on multiple datasets (CIFAR-10, CIFAR-100, Clothing-1M, CIFAR-N).

**Summary Of The Review:**

Although I think the openset noise problem in federated learning as defined by the authors is interesting and worth discussing in the research community. However, in the method design of FedPeer, the authors designed two stages “Privacy-preserving global label communication” and “Peer gradient updates”, in which both stages directly adopted existing methods and lacked novelty. Therefore, at this stage, I will vote for marginally below the acceptance threshold.

---

> ### Author Response · Authors · 2022-11-17
> **Response to Paper3039 Reviewer GPjn**
>
> We thank the reviewer for the constructive feedback! We hope to clarify the concerns and answer the questions below. Feel free to let us know if you have any further questions!
>
> 1. Thanks for your review. The term non-iid is too general for our cases. If the label distribution is not the same among the clients and every client can still observe the full label space, it is still a non-iid federated learning problem. Our setting is a subset of non-iid which only focuses on the scenario in that the client cannot view the full label space. Thus, we use the term ‘openset noisy label’.
>
> 2. Thanks for your effort in reading our paper. We are happy to find our example helps you understand our motivation. We will seriously consider your advice in the final version.
>
> 3. Thanks for your comments. First, FedPeer can protect the individual information and the proof is given in the Appendix. As for the leakage in distribution, the distribution information is usually relatively public. For example, we want to train a medical system to diagnose COVID-19 using CT scans from all over the world. The distribution of healthy people and patients is always public. However, the information of the individual is very private. Thus, the leakage of the distribution should not be a concern in practical usage.
>
> 4. Thanks for your comment. If our comprehension is correct, there is a misunderstanding about how we generate the openset dataset in federated learning. We will clarify this in the following. For the benchmark dataset, we apply the openset partition on the corrupted dataset rather than the clean dataset as we stated in Section 4.1 and Fig. 1. Thus, the true labels are not split evenly and it is harder for us to train a model.
>
> 5. Thanks for your comment. Your observation is a really interesting point. First, it can be found that although our method is not the best method in some settings, there is no large difference between our method and the best method. As for your question, one of the possible explanations may be that a higher noisy ratio will lead non-linear increment in the difficulty, the gap between 0.8 and 0.6 is not the same as that between 0.4 and 0.2. In highly corrupted cases, the application of label communication (DP process) can be viewed as another label corruption, which will enlarge this kind of uncertainty. Your point is worth investigating and can be the point of our future work.

---

> > ### Comment · Reviewer_GPjn · 2022-11-18
> > **Response to the Rebuttal**
> >
> > Thanks for the very detailed rebuttal, however, I maintain my score.
> >
> > The issue of distribution leakage may not be the focus of this work, but you cannot claim that distribution leakage should not be a concern in practical usage.
> >
> > In addition to the performance degradation of FedPeer under the high noise rate setting, we observe from Tables 5, 6, and 7 that FedPeer still does not show reliable performance when the recording standard is the last accuracy. FedPeer still has considerable potential for improvement.
> >
> > I think the main problem facing the article is the limited novelty. Although the authors raise an interesting and practical problem, 'openset noisy label', but the authors did not provide too much technical insights for the community in the method design. Peer loss is a commonly used label noise robust loss function proposed by Liu et al. In the label communication stage, FedPeer directly adopts the label differential privacy strategy proposed by Ghazi et al.

---

> > > ### Author Response · Authors · 2022-11-27
> > > **Response to Reviewer GPjn**
> > >
> > > We thank the reviewer for the constructive feedback! We hope to clarify the concerns and answer the questions below. Feel free to let us know if you have any further questions!
> > >
> > > 1. Thanks for your reviews. We acknowledge your point that the distribution leakage is one of the concerns of our work, which needs further improvement. However, so far to our best knowledge, there is no clear and realistic potential risk to expose the distribution leakage to the server in the real world. Could you please give a real example or mathematical proof to illustrate the risk?
> > >
> > > 2. Thanks for your careful reading of our appendix. We really appreciate it. We acknowledge that FedPeer is not always the best in terms of the last epoch standard. However, it should be noted that FedPeer is the **top 2 best methods** in most of the cases while the other baseline method cannot guarantee this in 2 public benchmarks and 4 datasets with practical noise.
> > > - Furthermore, As for the best epoch standard, FedPeer outperforms most of the baseline methods. It can also be found there is no large gap (The smallest is 0.03 and the largest is 1.60) between FedPeer and the best method when FedPeer is not the best.
> > > - In summary, although FedPeer cannot always show the best performance in these two standards, FedPeer can always give the users a **‘not bad’** performance which is a reliable performance from our perspective.
> > >
> > > 3. Thanks for your reviews. It should be noted that open-set noisy label is a difficult and realistic problem as the example we have given in the paper (Section 3.2) and the previous rebuttal and we are the first to formulate this question theoretically, make it individually private and solve it successfully without knowing the noisy ratio which is more close to the practical scenario. We want to claim that the usage of the negative labels is only one building block of the framework and peer loss is only one particular realization of the negative labels. **The framework is the main point of our paper.** Note peer loss cannot be directly implemented in our framework as we analyzed in Section 3.3 (the paragraph before Section 4). Moreover, no one makes peer loss work in FL settings and in openset noisy label. Our work will expand the horizon of noisy label and FL.

---

### Official Review · Reviewer_GxgA · 2022-10-25

**Confidence:** 3
**Correctness:** 3
**Technical Novelty And Significance:** 2
**Empirical Novelty And Significance:** 3
**Recommendation:** 5

**Clarity, Quality, Novelty And Reproducibility:**

The idea of applying peer loss in federated learning is interesting, however, the quality of this paper needs to improve. My main concerns are about the transmitted matrix initialization for real-world applications. The method is easy to reproduce.

**Strength And Weaknesses:**

Strength:
+ The idea in this work sounds interesting, which shares a global transmitted matrix in each clients to obtain the posterior label distribution for peer loss computation.

+ From the experiments, the proposed method indeed achieves some good results under label noises, compared with many other pipelines, such as FedAvg, FedProx etc.

Weakness:
- The initialization of transmitted matrix is important. For a real world federated learning system, how to initialize the transmitted matrix, since the data distributions for individual clients are unknown?

- In experiments setup, it is mentioned “The total communication round with the server R is 300 and differential privacy ε are 3.98, 5.98 and 3.95 for CIFAR-10, CIFAR-100 and Clothing-1M, respectively.”. How do those ε be selected?

- From Table 3, the performance is increased with increasing epsilon values. However, the best performance is obtained when 3.98 is set. 3.98 is very close to 4, which provides much worse performance. The results look unreasonable. Do you have any intuition or explanation for the results? Does the proposed method is not stable enough. If so, how do you select an epsilon value?

- In line 14 of Alg2, which loss do you use to measure the difference between predicted label and GT label? Cross entropy loss, or other loss? Does this loss function affect the results ?

- In Alg2, line 2,4,5 use y, while line 12-14 use Y. The same notation should be used.

- In the definition of T_{DP}, what is the meaning of K ? Parameter K is not explained.

**Summary Of The Paper:**

This paper presents a federated learning method under openset noisy labels. The idea of this paper is from the study of peer loss in centralized training with noisy labels. With global DP matrix, peer loss function can be computed and learn ML models with noisy models. To employ the peer loss in federated learning, the proposed method generates and broadcasts $T_{DP}$ to each clients to compute the estimated global peer loss. From the experiments and ablation studies, the proposed method performs better on datasets with synthetic label noises as well as real-world label noises.

**Summary Of The Review:**

This paper presents a new method for federated learning under noisy labels. The proposed method sounds interesting, however, there are several potential issues as discussed in the weakness section. As a result, I vote for "marginally below the acceptance threshold".

---

> ### Author Response · Authors · 2022-11-17
> **Response to Paper3039 Reviewer GxgA**
>
> We thank the reviewer for the constructive feedback! We hope to clarify the concerns and answer the questions below. Feel free to let us know if you have any further questions!
>
> 1. Thanks for your comment. We are puzzled about the initialization of the transmitted matrix mentioned in this comment. Does this refer to the noise transition matrix originally existing in the noisy dataset or the DP transition matrix? The DP transition matrix is initialized as a symmetric noise transition matrix following the equation above Theorem 1 (Page 6) and the noise level depends on the privacy level. Every client has the same DP matrix and it is independent of individual clients’ data distributions. The transition matrix of the original noisy dataset is synthesized as Sec. 5.2, which may be different for different clients. Please note our method does not need to estimate this matrix as we explained in Sec 3.3. For real-world datasets, we just use the noisy labels and do not need to synthesize any noise.
>
> 2. Thanks for your comment. The $\epsilon$ is calculated based on the Equation below Figure 1 on Page 6. The reason why we want to use 3.58 (we have corrected in this version), 5.98, and 3.95 is that we want to keep $\frac{e^{\epsilon}}{e^{\epsilon}+K-1}$ 0.2 corresponding to 10, 100, and 13 classes.
>
> 3. Thanks for your comment. The results are only 3 times on the CIFAR-10 dataset when the noisy ratio is 0.4 and the noise is random rather than symmetric. We increased the number of experiments to 10 and recorded every-time result shown below. The top 3 rows are the number we used to calculate in the original Table 3. The result valid our proof that there is no big difference among different $\epsilon$s. We think the minor fluctuations within 1-2 percent are the difference due to different initialization of the neural network, random label noise, and random selection of the clients in every round. In addition, we can also find that the worst result is much better than the baseline methods.
>
> |  epsilon  |      1     |      2     |      4     |      8     |     100    |    3.58    |
> |:---------:|:----------:|:----------:|:----------:|:----------:|:----------:|:----------:|
> |           |    67.78   |    67.74   |     70     |    70.64   |    70.03   |    75.49   |
> |           |    70.22   |    70.11   |    69.1    |    72.32   |    74.13   |    70.95   |
> |           |    70.12   |    69.6    |    70.99   |    70.33   |    76.37   |    71.88   |
> |           |    73.7    |    73.01   |    75.17   |    72.79   |    76.67   |    71.68   |
> |           |    74.78   |    71.83   |    74.23   |    73.95   |    74.32   |    71.9    |
> |           |    75.46   |    72.38   |    72.96   |    72.2    |    72.37   |    74.5    |
> |           |    72.1    |    71.35   |    73.47   |    75.44   |    73.37   |    71.13   |
> |           |    70.84   |    73.32   |    72.88   |    71.7    |    71.1    |    71.24   |
> |           |    75.22   |    74.28   |    72.15   |    76.17   |    76.88   |    72.45   |
> |           |    74.45   |    72.38   |    71.79   |    74.45   |    72.26   |    73.19   |
> | mean, std | 72.47±2.64 | 71.60±1.96 | 72.27±1.87 | 73.00±1.96 | 73.75±2.38 | 72.44±1.52 |
>
> 4. Thanks for your comment. The loss we used is the standard cross-entropy loss. But the structure of FedPeer or peer loss carries over other loss functions such as the hinge loss. However, in [1], they also find that cross-entropy loss works best in peer loss if the model is the neural network. Thus, we follow the findings of that paper in our ResNet.
>
> 5. Thanks for your comment. The $y$ in lines 2, 4, and 5 refer to the single data point and $Y$ is the label distribution. We use $y$ to better explain one-hot encoding. If it is clear to use a distribution like model prediction and loss calculation, we use $Y$. We appreciate your suggestions for clarity and we have revised it in this version, making it consistent with $y$.
>
> 6. Thanks for your comment. This $K$ is defined at the beginning of Section 3.1. The meaning of K is the size of the label space of the whole dataset. We have added another recall in T_{DP} to avoid ambiguity.
>
> [1] Liu, Y., & Guo, H. (2020, November). Peer loss functions: Learning from noisy labels without knowing noise rates. In International conference on machine learning (pp. 6226-6236). PMLR.

---

### Official Review · Reviewer_TkFx · 2022-10-25

**Confidence:** 3
**Correctness:** 3
**Technical Novelty And Significance:** 3
**Empirical Novelty And Significance:** 3
**Recommendation:** 5

**Clarity, Quality, Novelty And Reproducibility:**

$\textbf{Clarity}$: This manuscript is well-written and clearly structured, and the ideas are easy to follow and well-presented.

$\textbf{Quality}$: This manuscript is of good quality.

$\textbf{Novelty}$: While the core label-noise learning method is not novel [1], the authors still manage to make other non-trivial contributions, such as estimating open-set noisy label distributions under federated learning settings, and a clear formulation of the problem, pointing out the possible weakness of existing methods. I would say the novelty of this manuscript is enough for an ICLR paper.

$\textbf{Reproducibility}$: Authors released their implementation code with instructions for reproduction, I believe the results are reproducible.

[1] Liu, Y., & Guo, H. (2020, November). Peer loss functions: Learning from noisy labels without knowing noise rates. In International conference on machine learning (pp. 6226-6236). PMLR.

**Strength And Weaknesses:**

Pros:
1. The idea of using a transition matrix with a DP guarantee to generate DP label is interesting, I'm not sure if it is new (I'm assuming it is).
2. The research problem is clearly defined and well-supported by a concrete example, and the solution is strongly motivated.

Cons:
1. The authors should discuss how and why the DP label can still form a complete noisy label space. Based on my understanding, there is no guarantee that the DP label after flipping can still in fact form a complete noisy label space.
2. If the DP label transition matrix is generated by the server, wouldn't the server be able to invert the uploaded DP label with their respective T?
3. Authors' example on page 4 seems not correct, if a local T is estimated based on a label space without class 3, the third row of T should be all zeros. Specifically, based on commonly used anchor-point noisy posterior estimation [1,2], since there is no anchor point for class 3, the third row of estimated T should not exist (or all zeros).
4. The compared baseline methods should include some more recent methods and some state-of-the-art methods that also don't leverage the Transition Matrix for a fair comparison, such as Divide-Mix [3].

[1] Liu, T., & Tao, D. (2015). Classification with noisy labels by importance reweighting. IEEE Transactions on pattern analysis and machine intelligence, 38(3), 447-461.

[2] Xia, X., Liu, T., Han, B., Wang, N., Gong, M., Liu, H., ... & Sugiyama, M. (2020). Part-dependent label noise: Towards instance-dependent label noise. Advances in Neural Information Processing Systems, 33, 7597-7610.

[3] Li, J., Socher, R., & Hoi, S. C. (2020). Dividemix: Learning with noisy labels as semi-supervised learning. arXiv preprint arXiv:2002.07394.

**Summary Of The Paper:**

This manuscript studies an interesting and non-trivial problem, which is how to learn with open-set noisy labels under federated settings. The authors first give a very clear and easy-to-understand formulation of the federated open-set label noise problem, which is very useful. They then proposed a federated label-noise learning method by leveraging contrastive labels, and a Differential Privacy (DP)-preserving scheme to collect noisy labels without violating federated learning settings.

**Summary Of The Review:**

Overall, this manuscript studies an important and well-defined research problem and proposed a theoretically sound solution. However, the experiments should be more comprehensive to make their contribution more convincing, and whether their DP labels are privacy-preserving given DP transition matrix is debatable. I tend to reject this manuscript under its current version, but I am also willing to adjust my score upon a satisfactory and convincing rebuttal regarding my questions.

---

> ### Author Response · Authors · 2022-11-17
> **Response to Paper3039 Reviewer TkFx**
>
> We thank the reviewer for the constructive feedback! We hope to clarify the concerns and answer the questions below. Feel free to let us know if you have any further questions!
>
> 1. Thanks for your comment.
> * First, in FL, we generally assume all the clients share at least the same model structure. In this way, each client will know $K$, which is the output dimension of the model.
> * Second, we adopt the same *symmetric DP matrix* in every client. Therefore, as long as the total amount of data for each label class before DP (over all clients) is sufficient, the DP label after flipping still forms a full noise label space. For example, there are 100 points whose labels are all 1 and there are 10 classes in total, meaning the client can only access class 1. If the probability of label 1 flipped to the other classes is 0.05, the probability that there is no label 10 after flipping is $(1-0.05)^{100} = 0.00592053$.
> * Third, there is no requirement for every client to know the full space because we only need the distribution of the label to generate the artificial labels to calculate the 2nd element in peer loss as shown in Equation (5).
>
> 2. Thanks for your reviews. The DP label transition matrix is a noisy label transition matrix, which captures the **probabilistic** relationship between labels before and after flipping. Therefore, the server with the noisy label $\tilde{y}$ and noisy ratio $\epsilon$ cannot recover the individual true label $y$. It can only recover the distribution of $y$. For example, consider the case where the DP label is 2 and the transition probabilities are $\mathbb P(\tilde Y=2 | Y=1) = 0.2, \mathbb P(\tilde Y=2 | Y=2) = 0.8$. Due to $\mathbb P(\tilde Y=2) = \mathbb P(\tilde Y=2 | Y=1) \cdot \mathbb P(Y=1) + \mathbb P(\tilde Y=2 | Y=2) \cdot \mathbb P(Y=2)$, we cannot know the label before flipping for sure even though we know the corrupted label and the above transition probabilities exactly. Thus, it is impossible to recover the individual label.
>
> 3. Thanks for your reviews. We think the matrix you refer to is $T_{optEst}$ whose meaning is the optimal estimate of $T_{real}$. This $T_{optEst}$ is based on the knowledge of ground truth. For example, if we have 16 data points whose labels are 3. 1 of 16 points is flipped to 1 and 3 of 16 points are flipped to 2. Given the ground truth knowledge, if we can only observe class 1 and 2 among these noisy data,  the best estimation of the probability of flipping 3 to 1 is $1/4$ and flipping 3 to 2 is $3/4$ . The optimal estimator considered here does not rely on a particular estimation method. On the other hand, in FL, it is not clear whether the anchor-point estimator can find anchor points for class 3. Please note the answer is not trivial due to the model average with other clients which may have the knowledge of class 3.
> * As for your example, it is also valid. In the real application, there is no third column and the third row because the client has no access to the ground truth. We are very happy to add your example to the final version, making the paper more clear.
>
> 4. Thanks for your helpful comments. Divide-Mix is a powerful method in noisy label areas by applying mix-up methods. However, the data augmentation method is different from the way we usually use in Co-teaching, T-revision, etc. That is the reason why we did not include DivideMix. We ran DivideMix in FL settings, following the training process given in [1]. The implementation is added to the Appendix and some results of CIFAR-10 corrupted with random noise are as follows. We will finish more results soon.
>
> | noise_ratio |     0.2     |     0.4    |     0.6    |     0.8    |
> |:-----------:|:-----------:|:----------:|:----------:|:----------:|
> |    seed=1   |    72.52    |    61.30   |    48.13   |   21.78         |
> |   seed=10   |    68.15    |    57.56   |   44.77   |  25.29          |
> |   seed=100  |    70.25    |    58.87   |   53.84  |  25.79          |
> |  mean, std  |  70.31±2.19 | 59.24±1.90 |  48.91±4.58 |   24.28±2.19         |
> |     ours    | 82.15±0.24  | 72.69±1.57 | 59.06±1.38 | 27.55±1.49 |
>
>
> [1] Li, J., Socher, R., & Hoi, S. C. (2020). Dividemix: Learning with noisy labels as semi-supervised learning. arXiv preprint arXiv:2002.07394.

---

> > ### Comment · Reviewer_TkFx · 2022-11-18
> > **Good rebuttal, but not fully convinced**
> >
> > Thanks for the very detailed rebuttal, however, I'm not fully convinced by some of the authors' response.
> >
> > 1. While your example is very intuitive - the probability of forming a complete label space is large when n=100 and c=10. However, what if the sample size is much smaller (a very realistic scenario in FL), and the flip rate is larger? Then the probability of not having a full label space is considerable. I think there should be some theoretical guarantee for that, this point should be taken into consideration since the question of interest is open-set label learning.
> >
> > 2. I am aware that what the server can obtain is only the distribution of the labels, however, this also constitutes server information leakage. More specifically, knowing the underlying distribution of the labels can help the honest-but-curious server to better initiate gradient-inversion attacks [1].
> >
> > Lastly, authors are encouraged to release the implementation of the baseline methods for the reviewer to reproduce the experimental results.
> >
> > [1] Huang, Y., Gupta, S., Song, Z., Li, K., & Arora, S. (2021). Evaluating gradient inversion attacks and defenses in federated learning. Advances in Neural Information Processing Systems, 34, 7232-7241.

---

> > > ### Author Response · Authors · 2022-11-27
> > > **Response to Reviewer TkFx**
> > >
> > > We thank the reviewer for the constructive feedback! We hope to clarify the concerns and answer the questions below. Feel free to let us know if you have any further questions!
> > >
> > > 1. Thanks for your comment. It is very likely that the label after DP matrix cannot cover the full label space if the data points are very few. For example, if the client can only access 5 points for a 10-class classification task. However, as we mentioned in the first response (“Third, there is no requirement xxx” in the 3rd bullet point of the 1st part), the loss calculation in the local client does not need to know the full label space. The calculation only depends on the observed label and the artificial label sampled from the label distribution. Thus, there is no help for the client to know the data points belonging to all the classes.
> > >
> > > 2. Thanks for your reviews. We acknowledge that there is some issue with the distribution leakage. However, individual privacy is protected as shown in label communication in Section 4.1. We also check the paper you attached and the papers related [1, 2, 3]. However, we did not find a direct connection between the gradient-inversion attacks and the distribution leakage. Could you please clarify your point that the distribution leakage would lead to more severe gradient-inversion attacks in a more mathematical way?
> > >
> > > [1] Zhu, L., Liu, Z. and Han, S., 2019. Deep leakage from gradients. Advances in neural information processing systems, 32.
> > >
> > > [2] Zhao B, Mopuri KR, Bilen H. idlg: Improved deep leakage from gradients. arXiv preprint arXiv:2001.02610. 2020 Jan 8.
> > >
> > > [3] Geiping J, Bauermeister H, Dröge H, Moeller M. Inverting gradients-how easy is it to break privacy in federated learning?. Advances in Neural Information Processing Systems. 2020;33:16937-47.
> > >
> > > The implementation of dividemix in FL is https://drive.google.com/drive/folders/1Pogsgnqk-4VXBk_ni3Vxk5Jj3BKMN2UB?usp=sharing

---

> > > > ### Comment · Reviewer_TkFx · 2022-11-28
> > > > **Response to authors**
> > > >
> > > > Thank authors for your active participation in the discussion, regarding your rebuttal, here are my response:
> > > >
> > > > 1. Thank the authors for the clarification. However, this point is still somewhat unclear to me. If the client does not need the full label space, how do we aggregate their gradients? For instance, let's say that Client A only access label {0,1,2}, and Client B only access label {1,2,3}. The gradient each client compute are with respect to different labels, combining their gradients are ill-posed.
> > > >
> > > > 2. Thank the authors for the question, specifically, for the gradient-inversion attack, as mentioned in [1], a "strong assumption" is that the labels or the distribution of the labels are known. Under this assumption, existing methods can be better facilitated to invert the gradients. However, this assumption is usually hard to satisfy, since the labels are unknown. Yet, if the distribution of the label is leaked, we can use that as a regularization to better infer private information. For instance, using Zhu et al.'s work as an example [2], where we set the optimization objective as "minimizing the difference between true gradients and synthetic gradients". If the label distribution is unknown, there are two sets of unknown r.v.s that need to be recovered (X,Y), hence the solution might not be unique, and we might not be able to converge to the optimal. However, if the distribution of Y is known (i.e. P(Y)), we can then add a regularization term to minimize the KL divergence between P(Y) and P(Y_recovered). This gives us a tight constraint to the possible value range of the recovered X.
> > > >
> > > > This is simply an example, there are for sure better ways to do that. The point are "leaking the distribution of labels do not constitute information leakage is debatable" and "knowing the distribution of the labels will benefit gradient inversion attack from server end".
> > > >
> > > > [1] Huang, Y., Gupta, S., Song, Z., Li, K., & Arora, S. (2021). Evaluating gradient inversion attacks and defenses in federated learning. Advances in Neural Information Processing Systems, 34, 7232-7241.
> > > >
> > > > [2] Zhu, L., Liu, Z. and Han, S., 2019. Deep leakage from gradients. Advances in neural information processing systems, 32.

---

> > > > > ### Author Response · Authors · 2022-11-29
> > > > > **Response to Reviewer TkFx**
> > > > >
> > > > > We thank the reviewer for the constructive feedback! We hope to clarify the concerns and answer the questions below. Feel free to let us know if you have any further questions!
> > > > >
> > > > > 1. Thanks for your question. We would like to clarify that our framework is based on the assumption that the client does not need to **observe** the whole label space. In our first response, we used “know the full label space”, which might cause confusion. For example, there are four classes {0, 1, 2, 3} in the label space. The last layer of the client has 4 outputs since the clients know the full label space. However, the client does not need to **observe** the full label space, meaning that the client does not need to observe every instance from these 4 classes.
> > > > > Go back to your example. If Client A can access label {0, 1, 2} and client B can access label {1, 2, 3}. Then each client can just calculate gradients according to the (x,y) pairs in their local dataset. Following the above example, the model output would be a 4-dimensional vector, e.g, [0.6, 0.2, 0.1, 0.1], and calculating cross-entropy loss just needs to check the log of the $y^{th}$ element, e.g., $y=3$ and loss is $-\log{(0.1+\epsilon)}$. Note we have this $\epsilon$ term to avoid $\log{(0)}$. In our experiments, we let $\epsilon = 1e^{-7}$.
> > > > >
> > > > > 2. Thank you for your detailed example. We checked the papers following your comments but we find both papers did not explicitly demonstrate how the label distribution is used to facilitate an attack. We are not experts in this area but it is an interesting question to discuss.
> > > > > - Ref. [1] (Assumption 2) only assumed knowing private labels, which is not the label distribution. If we really miss some important statements, could you please point them out to clarify them?
> > > > > - It is an interesting idea to use add regularization terms according to the label distribution. However, Ref. [2] did not explicitly show this idea. We tried to understand your example that “If the label distribution is unknown, there are two sets of unknown r.v.s”. It is true that the (X,Y) pairs are not unique, but why are there only two sets of them? Besides, we agree that label distribution is extra information that may be used to help an attack, but it is not clear whether it is **sufficient** for a successful attack. In your example, minimizing the KL divergence between P(Y) and P(Y_recovered) may not guarantee a unique solution.

---

> > > > > > ### Comment · Reviewer_TkFx · 2022-12-06
> > > > > > **Response to authors**
> > > > > >
> > > > > > First, apologize for the typos, it should be "two r.v.s (X,Y)" instead of "two sets of r.v.s (X,Y)", thanks for noticing that.
> > > > > >
> > > > > > Thanks for authors' clarification on my first concern, which is well-addressed.
> > > > > >
> > > > > > And for the second concern, yes, you are right, the statement "revealing the distribution of labels will not **guarantee** an optimal recovery of $X$" is true. But, the statement that "revealing the distribution of labels will **guarantee a tighter constraint** to the optimal recovery of $X$" is also true. And, the second statement itself is enough to cause contradiction to the privacy concern within the FL structure. Also, as I have mentioned, sometimes just leaking the distribution of labels will tell a lot information [1].
> > > > > >
> > > > > > [1] Wainakh, A., Ventola, F., Müßig, T., Keim, J., Cordero, C. G., Zimmer, E., ... & Mühlhäuser, M. (2022). User-Level Label Leakage from Gradients in Federated Learning. Proceedings on Privacy Enhancing Technologies, 2022(2), 227-244.

---

> > > > > > > ### Author Response · Authors · 2022-12-07
> > > > > > > **Response to Reviewer TkFx**
> > > > > > >
> > > > > > > We thank the reviewer for providing these references and for your patient response. As in our previous response, we admit that the label distribution is additional information and it **may** be harmful to share it but it is not clear whether it is sufficient because there is **no rigorous proof**. In other words, you mentioned "distribution of labels will tell a lot of information", but this paper does not seem to either quantify the information or quantify how the label distribution helps attack. Could you please point it out in the literature more specifically like which sentence or which proof in case of our missing your point? We agree that it may be a concern but cannot claim it is **forbidden** if there is no direct proof.

---

### Official Review · Reviewer_21aY · 2022-10-27

**Confidence:** 3
**Correctness:** 3
**Technical Novelty And Significance:** 3
**Empirical Novelty And Significance:** 2
**Recommendation:** 5

**Clarity, Quality, Novelty And Reproducibility:**

Clarity:

The paper is well-written and very easy to follow.

Quality:

The paper considers federated learning with openset noisy labels and extends peer loss to the federated setting.

Novelty:

Given the existing work on peer loss in centralized label noise learning by Liu & Guo 2020 and Cheng et al., 2020, the novelty of the current paper is not strong.

Reproducibility: good

**Strength And Weaknesses:**

Strength

The paper considers federated learning with openset noisy labels, which may be practically useful. The proposed method is tested on both benchmark and real-world datasets and is shown to outperform baseline methods.

Weaknesses

The first concern is about novelty. As peer loss is well studied in centralized label noise learning by Liu & Guo 2020 and Cheng et al., 2020, the main contribution of the current paper is to extend it to the federated learning setting which seems quite straightforward.

Another concern is about the convergence of the proposed method. As the proposed method contains multiple steps, e.g., initialization with $T_{DP}$ generation, label communication, loss evaluation on noisy data, and peer gradient updates, the rate of openset noise and the selection of $\epsilon$ may inevitably affect training, how can we guarantee the convergence of proposed method?

The standard setup of federated learning is to train a model collectively from multiple parties without sharing each party's local data. The label communication step in FedPeer seems necessary to FedPeer but violates the standard federated setting. Could the authors provide some real-world examples of federated learning with openset noisy labels and justify label communication with the label privacy guarantee is enough?

**Summary Of The Paper:**

This paper studies federated learning where the local data on each client are non-iid in the sense that they may encounter openset noisy labels. To solve the problem, they extend the so-called peer loss which is studied in centralized label noise learning to the federated learning scenario and call the proposed method FedPeer. They also test the FedPeer method on both benchmark and real-world datasets.

**Summary Of The Review:**

The paper is generally interesting to me, but the novelty part is a concern and needs to be discussed carefully, e.g., what are the difficulties and challenges in the federated setting? Analysis of the convergence of the algorithm and more real-world examples are also needed.

---

> ### Author Response · Authors · 2022-11-17
> **Response to Paper3039 Reviewer 21aY**
>
> We thank the reviewer for the constructive feedback! We hope to clarify the concerns and answer the questions below. Feel free to let us know if you have any further questions!
>
> 1. We appreciate the reviews. Regarding your first concern. Peer loss is just the tool in our framework. The motivation of our paper is to solve the Openset Noisy Label Problem in Federated Learning (FL), a common but serious problem that has not been solved at this time. Our paper defines the format of the open-set problem seriously and explains why this problem cannot be solved by the existing methods using two toy examples in Section 2. Moreover, we design a framework of global label communication and peer gradient updates using differential privacy and peer loss. The framework successfully solved the problem in a safe and efficient way with a theoretical guarantee. Please note that **it is useless to solve the open-set problem if we just apply peer loss in the local clients without the global label communication and peer gradient updates.** Therefore, the focuses of the two papers are totally different and peer loss is only a part of $\textit{FedPeer}$. The whole structure is the novelty of this paper.
>
> 2. Thanks for your questions. We also have such concerns when we design FedPeer. We solved it as follows.
> * **DP Label Corruption:** For DP corruption, statistically, the label communication has no influence on the label distribution, i.e., $(T_{DP}^{\top})^{-1}T_{DP}^{\top}(\sum_{i=1}^{C}\tilde{\bf p}_n^{c})/C=\tilde{\bf p}$, which is highlighted in Section 4.1 in the newer version.
>
> * **Peer loss:** We proved in Theorem 2 that the gradient of every update of FedPeer is the same as centralized training with global peer loss. Then if the global peer loss can coverage, so do ours.
>
> * Therefore, the DP label corruption can be inverted and does not affect peer loss, and our local update has the same gradient as centralized training. Combined with the convergence of FedAvg, it is reasonable to believe the proposed method can converge.
>
> 3. Thanks for your comments. The idea of federated learning is to train the local data from every client and upload their model weight to the server so that data privacy can be protected. To protect local data privacy, we design FedPeer. The local client will corrupt the label first and then send the corrupted label to the client. After gathering the data, the client calculates the real data distribution using the $\epsilon$. Thus, there is no individual privacy leakage, which is in line with the idea of federated learning. The server and any other client have no idea about any information of where the data is from and to whom the data belongs. As for the distribution of the class, it is often public. For example, if we want to train a general model to diagnose COVID-19 using data from all over the world. The distribution of the infected and healthy people is public. Thus, we can use FedPeer to train a model regardless of the noisy ratio.

---

> > ### Comment · Reviewer_21aY · 2022-12-07
> > **Thanks for the response**
> >
> > I have read the authors' response and it's clear how the current FedPeer framework differs from the standard peer loss in centralized learning. However, I still have some concerns regarding the convergence of the proposed method which needs to be systemically studied in the future.

---

> > > ### Author Response · Authors · 2022-12-08
> > > **Response to Reviewer 21aY**
> > >
> > >  Thanks for your comprehension of our idea and our rebuttal. In Section B.2 of the Appendix, we have proved that in the federated learning scheme, our adaptation of peer loss has the same gradients as the centralized training. For the convergence, it can be found that the model is non-convex and the data is non-iid, therefore the plain convergence of FedAvg is **non-trivial** without sufficient assumptions, which is the **bottleneck** of giving a systemical proof. If we add some assumptions such as [1], we believe the convergence of FedPeer can be proved following their proof techniques. However, the focus of this paper is to define the openset noisy label problem and propose a new method or a framework to solve it. Your suggestion is very useful and we will seriously consider it as the direction for our next paper.
> > >
> > > [1] Li X, Huang K, Yang W, Wang S, Zhang Z. On the convergence of fedavg on non-iid data. arXiv preprint arXiv:1907.02189. 2019 Jul 4.

---

### Decision · Program_Chairs · 2023-01-20

**Decision:**

Reject

**Justification For Why Not Higher Score:**

N/A

**Justification For Why Not Lower Score:**

N/A

**Metareview: Summary, Strengths And Weaknesses:**

This paper was generally appreciated for practical applicability of the proposed approach (openset noisy labeled data and federated learning), but serious remarks have been also raised.
The main issues, among several requests of clarifications for some stages of the proposed method, regard the novelty aspects/incremental flavour, the need of improving experimental analysis - also in comparative terms - and to justify better the results, and, especially, the risk of information/privacy leakage with respect to the Federated Learning assumptions.
Authors rebuttal and discussion were raised over general and specific issues, but in the end all ratings remained below threshold, making this paper not acceptable for publication to ICLR 23.


**Summary Of Ac-Reviewer Meeting:**

N/A